# Retrospective Safety Evaluation of Combined Chlorambucil and Toceranib for the Treatment of Different Solid Tumours in Dogs

**DOI:** 10.3390/ani14233420

**Published:** 2024-11-26

**Authors:** Yuk-Yin Lai, Rodrigo Dos Santos Horta, Paola Valenti, Antonio Giuliano

**Affiliations:** 1Jockey Club College of Veterinary Medicine, City University of Hong Kong, Kowloon, Hong Kong, China; yukyinlai4-c@my.cityu.edu.hk; 2Department of Veterinary Medicine and Surgery, Veterinary School, Universidade Federal de Minas Gerais (UFMG), Belo Horizonte 31270-901, MG, Brazil; rodrigohvet@gmail.com; 3Clinica Veterinaria Malpensa AniCura, 21017 Samarate, VA, Italy; paola.valenti@anicura.it; 4CityU Veterinary Medical Centre, City University of Hong Kong, Kowloon, Hong Kong, China; 5Department of Veterinary Clinical Sciences, Jockey Club College of Veterinary Medicine, City University of Hong Kong, Kowloon, Hong Kong, China

**Keywords:** Chlorambucil, Toceranib, combined, safety, tumours, dogs

## Abstract

Chlorambucil and Toceranib are generally safe for canine chemotherapy, respectively, with rare severe adverse events, but their combined use has not been studied. This study aimed to evaluate the safety profile and response rate of combining Chlorambucil and Toceranib in canine solid tumours. A retrospective analysis was conducted on thirty-eight dogs treated with this combination from two referral hospitals between January 2020 and September 2024. Chlorambucil was administered as a 2 mg tablet, with varying schedules based on dog size, resulting in a median dose intensity of 15.1 mg/m^2^ per week. Toceranib was given at a median dose of 2.5 mg/kg on a Monday–Wednesday–Friday schedule, with adjustments being made for co-morbidities and adverse events. The combination therapy achieved a clinical benefit rate of 55.3%. Mild gastrointestinal adverse events were the most common, occurring in 39.5% of cases, followed by UPC elevation in 15.8% of the cases, while hematological and biochemistry adverse events occurred in 13.2% of the cases each. No dogs discontinued treatment due to adverse events, indicating the combination is safe and well tolerated.

## 1. Introduction

Chlorambucil (Leukeran, or 4-[bis(2-chlorethyl)amino]benzenebutanoic acid) is an aromatic derivative of nitrogen mustard, which was initially used for the treatment of chronic lymphocytic leukemia (CLL) in people [1]. In the late 1960s, Chlorambucil became the first drug incorporated into canine chemotherapy protocol in combination with prednisolone [2].

As an alkylating agent, Chlorambucil enters cells by passive diffusion and via DNA alkylation, and, through targeted binding to DNA N7 guanine sites, it causes the formation of intrastrand or interstrand cross-linkages involving DNA bases or DNA–protein interactions [3,4,5,6,7]. In vivo studies demonstrated DNA alkylation, inducing apoptosis with a P53-dependent pathway that leads to a subsequent decrease in Bcl-2 and an increase in Bax, contributing to its anti-cancer properties [3,4,5,6]. Chlorambucil treatment has been shown to reduce the number and function of CD25+, CD4+ and Foxp3+ regulatory T cells in the peripheral blood [8,9]. It also inhibits endothelial cell proliferation and migration, and it is responsible for angiogenesis suppression [10]. Chlorambucil has been used as metronomic chemotherapy treatment in prospective clinical trials on dogs with various carcinomas, mast cell tumours (MCTs), and sarcomas as others, in which the metronomic treatment can provide a potential benefit with a low toxicity profile over the conventional maximum tolerated dose chemotherapy [11,12,13,14].

Chlorambucil has been extensively used in veterinary medicine as a primary treatment option for CLL and low-grade/indolent lymphoma [15,16,17,18,19]. It showed favourable tolerability and beneficial outcomes in dogs with various solid malignancies, such as glioma, urinary bladder transitional cell carcinoma, and MCTs [13,15,20,21,22].

Long-term administration of Chlorambucil at dosages above 3–4 mg/m^2^ per day may result in hematological toxicity and chronic myelosuppression, specifically neutropenia and thrombocytopenia (TCP) [23,24]. Hematological adverse events (AEs), typically ranging from mild to moderate, are often reversible; however, in rare instances, they may lead to irreversible and progressively severe TCP, even following medication discontinuation [23,24]. In the context of treating CLL, a recommended therapeutic regimen entails an initial daily dosage of 6 mg/m^2^ for 1–2 weeks, followed by a reduced daily dosage of 3 mg/m^2^ [17]. In another study, Chlorambucil administration at doses higher than 6 mg/m^2^ did not result in a higher clinical response but was shown to increase the risks of experiencing AEs [23]. Seizures have been reported sporadically following Chlorambucil treatment in both veterinary and human medicine, especially in pediatric patients with nephrotic syndrome, but studies have also revealed minimal or undetectable levels of Chlorambucil in the cerebrospinal fluid of treated dogs, along with a short half-life, indicating a high safety threshold at 4 mg/m^2^ SID [21,25,26,27,28].

Fanconi syndrome is characterized by aminoaciduria and glucosuria despite normoglycemia, and it was also recently described in four cats during treatment with Chlorambucil; however, there are no reports regarding its use in dogs [29].

Chlorambucil is overall considered well tolerated in dogs, even when administered for long periods of time at a recommended safe dosage of no more than 3–4 mg/m^2^ SID [21,23,24,25,26,27,30].

Toceranib (Palladia, SU11654) is an oral tyrosine kinase inhibitor (TKI) developed by Pfizer Animal Health Inc. (New York City, NY, USA), which was licenced by the Food and Drug Administration in 2009 for the treatment of unresectable or recurrent Patnaik grade II to III canine cutaneous MCTs with or without regional lymph node involvement [31,32,33].

As a multi-kinase inhibitor, Toceranib allosterically binds to the ATP binding site of various kinases, especially KIT, vascular endothelial growth factor receptors (VEGFRs), and platelet-derived growth factor receptors (PDGFRs), which consequently impedes phosphorylation kinase activation and cellular signal transduction for gene transcription [32,34,35,36,37,38,39,40,41,42,43]. In addition, Toceranib, used in combination with metronomic chemotherapy, was also shown to induce a significant synergistic effect on regulatory T cell depletion [44,45].

Apart from the licenced treatment for MCTs, Toceranib also demonstrated some clinical benefits in controlling various solid tumours, including head-and-neck carcinomas, thyroid carcinomas, insulinomas, anal gland anal sac adenocarcinomas (AGASACAs), heart base tumours, and gastrointestinal stromal tumours (GISTs) [34,46,47,48,49,50,51,52,53,54].

The initially established label dose for oral Toceranib administration was 3.25 mg/kg as administered EOD, which demonstrated significant efficacy with increased survival time in dogs diagnosed with an MCT [33,50]. However, mild to moderate hematological and gastrointestinal AEs could be commonly experienced, with the most described AEs including anorexia, diarrhea, vomiting, anemia, and neutropenia [32,51,55,56,57].

One potential mechanism for gastrointestinal toxicity in Toceranib could be attributable to the inhibition of the KIT protein on the interstitial cells of Cajal, which thereby induces gastrointestinal hypomotility, while the inhibition of KIT in bone marrow stem cells could be responsible for hematological toxicity [58,59,60,61]. Renal toxicity and proteinuria could be the result of targeting VEGFRs and PDGFRs, which can also lead to local thrombosis and endothelial damages [62,63,64,65]. Multiple research projects demonstrated in vivo on canine and mouse models that, with the reduced dosage at 2.4–2.9 mg/kg, Toceranib, could remain at an effective plasma concentration for inhibiting target RTKs [39,41,57]. Subsequent studies investigating the administration protocol of Toceranib at a dosage of 2.8 mg/kg on a Monday–Wednesday–Friday schedule unveiled an improved safety profile [46,57,66].

The main aim of this study was to retrospectively evaluate the safety of the combination of conventional metronomic chemotherapy, Chlorambucil, with targeted therapy, Toceranib, which could have potential synergistic effects on different solid tumours. Additionally, the secondary aim was to document the preliminary response rate of this combined chemotherapy protocol across various canine cancer types.

## 2. Materials and Methods

Medical records of dogs from two referral hospitals, treated with the combination of Chlorambucil and Toceranib from January 2020 to September 2024 were retrospectively evaluated. All dogs with a confirmed diagnosis of a malignant solid tumour that received concurrent administration of Chlorambucil Excella GmbH & Co. KG (Feucht, Germany) and Toceranib Pfizer Animal Health Inc. (New York City, NY, USA) has a combined chemotherapeutic treatment was included. Tumours at any advanced stage, and primary or recurrent tumours after surgery, were all included. Dogs were excluded if they did not have an adequate follow-up, i.e., did not return to the hospital for clinical assessment and blood work after starting the combined therapy.

The data obtained from the clinical record included sex with neuter status, breed, age, tumour type and stage, start and end dates of the combined chemotherapy (specifying if the treatment was first-line or second-line treatment), body weight at the beginning and end of the combined chemotherapy, clinical and objective treatment response, AE categorized as gastrointestinal, hematological, renal or others, dosage of each chemotherapeutic agent, survival time, and reason of death or euthanasia when known.

All the dogs were treated with a combination of Toceranib and Chlorambucil at various doses and schedules. A commercially available 2 mg tablet of Chlorambucil was used in all dogs, so the schedule of administration varied mainly based on the size of the dogs from daily to biweekly. Individual adjustments were also made during treatment to the administration schedule and dosages of the treatment for each single dog at the clinician’s discretion.

AEs were classified and graded according to the grading system from the Veterinary Cooperative Oncology Group Common Terminology Criteria for Adverse Events (VCOG-CTCAE v2) [67]. The response rate was assessed following the Veterinary Cooperative Oncology group’s RECIST response guideline for solid tumours [68]. Despite the retrospective nature of this study, drug dosages, re-staging, and evaluation of response were consistent overall, as all the cases were seen by the same clinician at the same institution. Overall, the re-staging and response assessment was performed every 3–6 weeks, with the exact frequency for individuals being adjusted in conjunction with the clinician’s decision, tumour type and stage, and/or owner’s wish. The response to treatment was assessed by physical examination, calliper measurement, and/or imaging when appropriate, regarding the nature and location of the solid tumours.

Only dogs with objectively measurable gross disease were assessed for response. The individual response towards treatment was defined as complete response (CR), with complete resolution from disease being based on clinical and imaging evidence; partial response (PR), with reduction in tumour diameter by no less than 30% and no new development of lesions; stable disease (SD), with changes in tumour diameter no more than ±20% and no new development of lesions; or progressive disease (PD), with an increase in tumour diameter more than 20% or showing new development of lesions. Overall response rate (ORR) was calculated by summation of dogs achieving CR and PR, while clinical benefit rate (CBR) was calculated by summation of dogs achieving CR, PR, and SD for at least six weeks. Survival time was calculated in days between the date of the cancer being diagnosed and the date of death, including euthanasia, of the dog. Due to the small number of cases, statistics were only descriptive. The progression-free survival (PFS) of an individual was calculated as the number of days between the date of the first administration of chemotherapy and the date of disease progression. Median survival time (MST) was obtained with a Kaplan–Meier product-limit method. Meanwhile, dogs who were lost at follow-up, or were alive at the time of analysis, were censored. Statistics were performed with IBM SPSS Statistics. Considering the small sample size and the heterogenicity of the dogs treated for various cancer types, MST and PFS were calculated only for specific cancer types with the highest number of cases.

## 3. Results

Thirty-eight dogs met the inclusion criteria, with nineteen males and nineteen females. Among the dogs included in our study, nineteen of them were non-resectable tumour cases, and the other nine cases had surgery before starting the combination treatment. None of the cases had received radiotherapy. The most common breed was poodle (*n* = 11), followed by other breeds including schnauzer, Bichon Frise, and others, as summarized in Table 1. The median age of the dogs when diagnosed with cancer was 11 years old, with a mean of 10.9 years old (range: 5–15.9 years). There were twenty-three dogs diagnosed with carcinoma, eight diagnosed with an MCT, six diagnosed with sarcoma, and one diagnosed with a GIST. Of the twenty-three carcinoma cases, there were two thyroid carcinomas, four lung carcinomas, three nasal carcinomas, one frontal sinus carcinoma, seven mammary carcinomas, one lingual SCC, one prostatic carcinoma, two AGASACAs, one hepatocellular carcinoma, and one clear cell renal carcinoma. There were three cases of stage II mammary carcinoma, one case of stage III mammary carcinoma, and two cases of stage V mammary carcinoma. Regarding the two stage II cases, they were recurred tumours after surgery, with widespread satellite cutaneous nodules, hence the starting of chemotherapy. The third stage II case also had a low-grade T cell lymphoma. This dog was on Chlorambucil for low-grade T cell lymphoma initially, and Toceranib was added when it developed metastatic mammary carcinoma. For the stage III case, the owner opted for the combined therapy of Chlorambucil and Toceranib as the initial treatment. For the two stage V cases, they had lung metastases and were only treated with Chlorambucil and Toceranib. The case with stage IV nasal carcinoma had developed brain invasion, with widespread musculoskeletal metastasis. Of the eight MCT cases, six were high-grade MCT and two were low-grade MCT, staged between III and IV, while four cases were recurred cancer. Of the six sarcoma cases, there were three hemangiosarcoma cases (one classified as stage II), one liposarcoma, one recurred fibrosarcoma, and one occipital region sarcoma with brain invasion. Data are summarized in Table 1.

Chlorambucil was administered using the 2 mg tablet at various schedules depending on the size of the dog, with a median dose being equivalent to 2.1 mg/m^2^ daily (range 1.4–5.3 mg/m^2^) and a median dose intensity of 15.1 mg/m^2^ (range 9.5–37.2 mg/m^2^) per week. One dog received a 2 mg tablet weekly, yielding an actual dosage of 1.4 mg/m^2^, considerably below the average dose. This was due to the limitations of the small-sized dog and only the 2 mg tablet formulation being available.

Toceranib was administered at a median of 2.5 mg/kg (range 1.7–3 mg/kg) three times a week. There were two dogs administered with a dosage < 2.0 mg/kg due to already present diarrhea from previous treatment. One dog received 2.6 mg/kg of Toceranib monotherapy as first-line treatment and developed G2 diarrhea. As a result, the treatment was switched to the Chlorambucil and Toceranib combination for second-line treatment, with the Toceranib dosage reduced to 1.7 mg/kg. Another dog received 2.6 mg/kg of Toceranib monotherapy with 20 mg tablet as first-line treatment but developed G2 diarrhea and required a dose reduction. Consequently, the dog changed to the Chlorambucil and Toceranib combination for second-line treatment, with the Toceranib dosage being reduced to 1.9 mg/kg.

The dosages of Chlorambucil and Toceranib for each dog were summarized in Table 1. Adjustments to dosage and schedule were made regarding the individual clinical case and if any AEs were encountered. For dogs experiencing gastrointestinal AE, the drug combination was temporarily discontinued until improvement was observed, followed by a reduction in the dose of Toceranib.

### 3.1. Adverse Events

The chemotherapy protocol was well tolerated by the majority of dogs included in this study, with only occasional and mild AEs, mainly gastrointestinal and hematological, with 47.4% dogs (*n* = 18) experiencing some forms of or a combination of AEs. Among all the categories of AEs, 86.4% were limited to G1 and G2 and had shown improvement after reduction in drug dose or administration schedule. The summary of AEs is shown below in Table 2.

Gastrointestinal AEs were the most common, as experienced by 39.5% of the dogs (*n* = 15) under combined treatment, including diarrhea, inappetence/anorexia, and vomiting. In terms of hematological abnormalities, five dogs (13.2%) experienced neutropenia and/or TCP. Biochemistry AEs were experienced by five dogs (13.2%). In terms of renal abnormalities, six dogs (15.8%) were found to have proteinuria and a high urine/protein creatinine (UPC) ratio. All the AEs were mainly G1–2 and resolved with dosage adjustment. The AEs in dogs receiving the combination of Chlorambucil and Toceranib is summarized in Table 2 below.

One dog initially started with 2 mg Chlorambucil weekly (9.5 mg × m^2^ weekly; 1.4 mg × m^2^ daily) and 3.0 mg/kg Toceranib triweekly, experiencing G2 anorexia and G3 UPC elevation. Another dog that initially started with 2 mg Chlorambucil biweekly (12.0 mg × m^2^ weekly; 1.7 mg × m^2^ daily) and 2.5 mg/kg Toceranib triweekly later experienced G1 TCP, G1 neutropenia, G2 diarrhea, and G3 anorexia. The third dog initially started with 2 mg Chlorambucil daily (23.9 mg × m^2^ weekly; 3.4 mg × m^2^ daily) and 2.1 mg/kg Toceranib triweekly but experienced G2 diarrhea and G1 anorexia. One dog was initially administered with 2 mg Chlorambucil biweekly (10.8 mg × m^2^ weekly; 1.5 mg × m^2^ daily) and 2.7 mg/kg Toceranib triweekly and then experienced G1 neutropenia, G1 anorexia, and G2 ALT elevation. The fifth dog, treated with 4 mg Chlorambucil six times a week (26.7 mg × m^2^ weekly; 3.8 mg × m^2^ daily) and 2.75 mg/kg Toceranib triweekly, experienced G1 anorexia. Among all the five dogs, the AEs were resolved after dose or frequency adjustments, and no dogs stopped the treatment due to any severe AEs.

### 3.2. Responses and Survival

Of the thirty-eight dogs included, all were assessed for objective responses, 55.3% achieved clinical benefit (CBR; *n* = 21), and 10.5% showed objective response (ORR; *n* = 4). Within the dogs demonstrating objective response, there was one MCT dog achieving CR and three carcinoma dogs achieving PR. Regarding the other twelve dogs achieving SD, there were fourteen carcinoma, one MCT, and one sarcoma.

Of the twenty-three carcinoma dogs, 73.9% achieved clinical benefit (*n* = 17). There was one lung carcinoma and one lingual carcinoma dog showing PR. Meanwhile, there were three lung carcinomas, one nasal carcinoma, one frontal sinus carcinoma, five mammary carcinomas, one prostatic carcinoma, two AGASACAs, one hepatocellular carcinoma, and one clear cell renal carcinoma that achieved SD. Of the eight MCT dogs, 25% achieved clinical benefit (*n* = 2), with one achieving CR and one achieving SD. Of the six sarcoma dogs, 16.7% achieved clinical benefit with SD (*n* = 1, hemangiosarcoma).

Of all the included dogs, 73.7% of dogs (*n* = 28) were treated with Chlorambucil and Toceranib as the second-line treatment, with thirteen dogs having only Toceranib as prior treatment and three dogs having only Chlorambucil as prior treatment. On the one hand, all the MCT dogs showing clinical response received the Chlorambucil and Toceranib combination as a second-line treatment. On the other hand, all the sarcoma dogs showing clinical response received the combination as a first-line treatment. Meanwhile, of the sixteen carcinoma cases showing clinical benefits, 18.8% received the combined therapy as first-line treatment (*n* = 3) while 81.3% received it as second-line treatment (*n* = 13).

Regarding all the thirty-eight dogs, the median follow-up was 211 days. At the time of writing, five dogs were lost to follow-up, while two dogs were still alive. Meanwhile, thirty-six dogs were available for the PFS calculation, and the other two dogs were censored as PFS was not provided. Of the thirty-six dogs that died, sixteen (44.4%) were euthanized, fourteen (38.9%) were lost to cancer-related death, one (2.8%) was lost to spontaneous death of unknown origin, and five (13.9%) were lost to follow-up.

The overall MST of all dogs was 259 days, while the MST for dogs receiving the combination of first-line and second-line treatment were 114 days and 351 days, respectively; Figure 1. The overall MST for dogs with carcinoma, MCT, and sarcoma were 315 days, 209 days, and 167 days, respectively.

The overall median PFS of all analyzed dogs was 45.5 days, while the median PFS for dogs receiving the combination of first-line and second-line treatment were 32.5 days and 48 days, respectively. The overall median PFS for dogs with carcinoma, MCT, and sarcoma were 49 days, 19 days, and 20.5 days respectively.

A pattern for all dogs having a longer MST was observed for those receiving the combination of second-line therapy versus first-line therapy (*p* < 0.001); however, due to the small number of dogs per cancer type group, these could not be evaluated statistically.

## 4. Discussion

The safety of combining Toceranib with Chlorambucil has never been investigated. The combination appears to be generally safe, with a toxicity comparable to Toceranib alone. In our study, most AEs were mild. All dogs experiencing AEs could be resolved successfully through dose reduction or schedule adjustments. None of the dogs from our study developed severe AEs that required cessation of the combined therapy and not treatment-related hospitalization or death.

Gastrointestinal AEs were the most common, occurring in 39.5% of our cases, with only two dogs experiencing G3 anorexia; the rest experienced G1 to G2 AEs, which were resolved after a Toceranib dose reduction. In a study utilizing Toceranib at a median dose of 2.5 mg/kg EOD, gastrointestinal AEs were observed, with 30.8% of cases (*n* = 15) showing G1 to G3 diarrhea, 20.5% of the cases (*n* = 8) showing anorexia, and 5.13% (*n* = 2) showing vomiting, of which 10.3% (*n* = 4) required a dose reduction [55]. Similarly, another study using Toceranib at 2.75 mg/kg EOD in dogs also reported gastrointestinal AEs as the most common, accounting for 53% of all AEs, including most commonly G1 to G3 diarrhea, followed by G1 to G2 vomiting and G1 to G3 anorexia [56]. A study employing Chlorambucil at 4 mg/m^2^ EOD showed 11.1% gastrointestinal toxicity (*n* = 4), with mainly G1 to G2 [14]. Furthermore, for dogs treated with Chlorambucil at 4 mg/m^2^, 6 mg/m^2^ and 8 mg/m^2^ SID as a comparison, those receiving 4 mg/m^2^ were reported to have less severity in terms of gastrointestinal AEs, which further supports that using the combined therapy with Chlorambucil applied as a metronomic at a lower dose could reduce the incidence and severity of toxicity [23]. In our study, we found gastrointestinal AEs, including mainly G1 to G2 anorexia, vomiting and diarrhea, which is similar to Toceranib monotherapy, suggesting that adding Chlorambucil at an average dose of 2 mg/m^2^ daily to Toceranib does not significantly increase gastrointestinal AEs.

Hematological AEs were the second most common from our study and involved 13.2% of cases experiencing neutropenia and/or TCP; most of them were G1, with only one case of G3 TCP. More neutropenia events were noticed than TCP in our study. In the study comparing the tolerability of metronomic Chlorambucil at 4 mg/m^2^, 6 mg/m^2^, and 8 mg/m^2^ SID in dogs, TCP and neutropenia were also noticed [23]. TCP was more frequent, with 28% of dogs (*n* = 78) experiencing the severity of G1 to G2 [23]. Additionally, 19% of the dogs (*n* = 15) required discontinuation of Chlorambucil due to TCP (24). It was indicated that, the higher the dosage of Chlorambucil utilized, the earlier the TCP would develop in the dog [23]. Sixty percent of the dogs (*n* = 3) receiving 8 mg/m^2^ Chlorambucil developed persistent TCP until death or loss of follow-up [23]. Neutropenia was observed in 6.5% of the dogs, limited to also G1 to G2 [23]. A study evaluating AEs of Toceranib in dogs revealed 50% of G1 neutropenia (*n* = 20), 2.5% of G2 neutropenia (*n* = 1), and 5% of G1 thrombocytopenia (*n* = 2), while the neutropenia cases were either resolved or did not worsen [57]. Neutropenia was also reported in another study in dogs treated with Toceranib as a monotherapy, with 21% (*n* = 6) experiencing certain grades of neutropenia [55]. Therefore, Toceranib in our combined therapy contributed to only mild haematological AEs, and the combined therapy had a similar hematological safety profile as either of the drugs utilized as monotherapy.

Proteinuria was experienced by 15.8% of dogs in our study, with G1 to G3 UPC elevation. This is similar to what has been published for treatment with Toceranib in monotherapy. Dogs treated with Toceranib as a monotherapy against solid tumours were also reported to have 10% of the dogs experiencing G1 protein-losing nephropathy [57]. In a study evaluating dogs receiving Toceranib, 24.5% of dogs (*n* = 12) that were not proteinuric before treatment developed proteinuria, which included 14.3% G1 (*n* = 7), 2.9% G2 (*n* = 1), and 8.2% G3 (*n* = 4) [69]. Among these dogs treated by Toceranib with concurrent NSAID or glucocorticoid, the median time to develop proteinuria was 47 days (range 19–626) and 69 days (range 15–212), respectively [69]. On the one hand, a study revealed 21% (n = 6) of dogs developing proteinuria within the treatment group of Toceranib [70]. However, there was no significant difference detected upon the median UPC in dogs on day 0 and day 14 of Toceranib treatment, suggesting proteinuria could be concurrent with the neoplastic processes [70]. Meanwhile, among normotensive dogs prior to Toceranib treatment in the same study, 37% of them (*n* = 11) developed systemic hypertension, which may, on the other hand, indicate being predisposed to proteinuria development after treatment [70].

Biochemistry AEs were reported in 13.2% of our dogs with G2 to G3 ALT elevation and G2 creatinine elevation. Similarly, it was reported that 27.6% of dogs (*n* = 40) and 13.8% of dogs (*n* = 20) developed ALT and creatinine elevation, respectively, while the dogs were administered with 3.25 mg/kg Toceranib EOD [32]. In our study, only 1.4% (*n* = 2) dogs experienced G3 to G4 ALT elevation, and another 1.4% (n = 2) experienced G3 to G4 creatinine elevation. Another study assessing toxicities in Toceranib-treated dogs reported 22.5% of the dogs (*n* = 9) experiencing ALT elevation, with 17.5% G1 (*n* = 7), 2.5% G3 (*n* = 1), and 2.5% G4 (*n* = 1), while they were treated with 2.5 mg/kg or 2.75 mg/kg Toceranib EOD [57]. Furthermore, there was another AE evaluation regarding Chlorambucil in cats with small cell lymphoma which revealed variable hepatotoxicity among the treated dogs, involving 24.5% having ALT elevation (*n* = 13), with improvement being shown after discontinuing Chlorambucil treatment [71]. However, there were limited reports of ALT elevation in dogs receiving Chlorambucil treatment. It is therefore likely that the biochemistry AEs are more attributable to Toceranib alone.

In addition to the primary objective of investigating AEs associated with combined therapy, we also reported response rates and survival for the main cancer types. Among the thirty-eight dogs treated with the Chlorambucil and Toceranib combined therapy, four dogs benefited from a long survival of greater than two years. Two of them were diagnosed with stage II lung carcinoma, which achieved SD with 1078 days of ST (PFS 262 days) and PR with 897 days of ST (PFS 382 days), respectively. One dog diagnosed with stage IIIb AGASACA was treated with combined therapy after debulking surgery of the iliac metastasis of the tumour, achieving SD with 725 days of ST (PFS 283 days). One dog diagnosed with stage IIIa MCT achieved CR for 1.5 years and 1124 days of ST (PFS 537 days). These dogs all received the combination of Chlorambucil and Toceranib as a second-line therapy. Among the dogs receiving the Chlorambucil and Toceranib combination as first-line therapy (n = 10), 50% of them achieved clinical benefits as SD. One dog who was diagnosed with splenic hemangiosarcoma achieved 111 days of PFS with 184 days of ST. Notably, hemangiosarcoma, in general, is highly progressive, with ST being approximately 90–180 days [72]. Additionally, in another study, none of the hemangiosarcoma dogs (n = 3) receiving Toceranib as monotherapy could achieve SD for over 70 days [40].

Surprisingly, dogs receiving the combined chemotherapy as second-line therapy had longer survival compared to dogs receiving the treatment as first-line therapy. This is likely caused by the small number of different cancer types cases and potentially a selection bias, as most dogs receiving the combined chemotherapy as first-line therapy were advanced stage/post-surgical recurrent cancers, and/or cancer with a more aggressive biological behaviour, i.e., hemangiosarcoma.

Chlorambucil and Toceranib combination chemotherapy is likely to be well tolerated in dogs, and this combination could have potential synergistic effects in the treatment of certain types of cancer. One potential synergistic effect could be related to the anti-neoangiogenic effect of both drugs, which is achieved by the endothelial vessel cytotoxicity of continuous low-dose Chlorambucil and by inhibition of VEGFRs and PDGFRs by Toceranib [7,30,73]. It is shown that Chlorambucil could inhibit direct sprout formation in an in vivo model with vascular occlusion [73]. Chlorambucil could also induce selective cytotoxicity in human-immortalized endothelial cells, impeding the viability of endothelial cells [7]. Inhibition of VEGFRs and PDGFRs by Toceranib could inhibit endothelial migration, proliferation, and neo-angiogenesis in various cancer types [37,38,74]. Additionally, Toceranib might impede the synergistic effects on VEGF by inhibiting the fibroblast growth factor receptor in angiogenesis [37]. The combination of angiogenesis inhibitors with other drugs in a specific temporal sequence has been proposed to synergistically augment therapeutic outcomes by promoting pericyte coverage and facilitating vessel normalization, ultimately impeding metastasis, and enhancing the tumour microenvironment [75]. Hence, by improving the delivery of oxygen and drugs, this approach mitigates therapeutic resistance induced by hypoxia [75]. The combined therapy approach involving Chlorambucil and Toceranib could potentially enhance the penetration of the anti-cancer drug Chlorambucil synergistically, as Toceranib promotes the stabilization of blood vessels [37,38,74].

Metronomic chemotherapy is known for its anti-angiogenic effect but also for restoring anti-cancer immunity. A study involving dogs with soft-tissue sarcomas revealed an elevated amount of regulatory T cells in dogs with cancer compared with a control group and that the use of metronomic therapy resulted in a decrease in regulatory T cells [44]. Another study also showed the synergic effect of the association of Toceranib and Cyclophosphamide in dogs with various types of cancer in which Toceranib significantly decreased the number and percentage of regulatory T cells in the peripheral blood after two weeks of treatment [45].

The main limitations of this study are the retrospective nature of the data collection, the different types and stages of cancer investigated, and the small number of cases. It is notable that, due to the retrospective nature of the study, there was not a standard dose for Chlorambucil and Toceranib. However, at the range of doses investigated, the combination appeared relatively safe and opened the possibility of performing a safe prospective study, with dose escalations possibly starting at 2 mg × m^2^ daily. The main aim of this study was not to evaluate efficacy, so we cannot give any conclusion on efficacy. However, some of the responses reported were promising, but a larger prospective study on specific types of cancers with escalating dosages of Chlorambucil is needed.

## 5. Conclusions

The safety of combining Toceranib phosphate with Chlorambucil has never been investigated. In our retrospective study, Chlorambucil was administered at the median dose intensity of 15.1 mg/m^2^ (range 9.5–37.2 mg/m^2^) weekly and Toceranib was given at the median dosage of 2.5 mg/kg (range 1.7–3 mg/m^2^) on a Monday–Wednesday–Friday schedule. The combination appears safe, with toxicity comparable to previously published studies on Toceranib alone. Most of the dogs experienced mild AEs, between G1 to G2, with no G3 or above AEs, and no dogs were required to permanently discontinue the drug combination. A synergistic effect of the two drugs in combination is possible and could improve outcome in certain cancer types. These findings may support the potential of this combination approach as an effective and well tolerated treatment strategy for diverse solid tumours in dogs. A large prospective clinical trial is needed to assess the safety and, especially, the efficacy of this drug combination in regard to a specific type of cancer.

## Figures and Tables

**Figure 1 animals-14-03420-f001:**
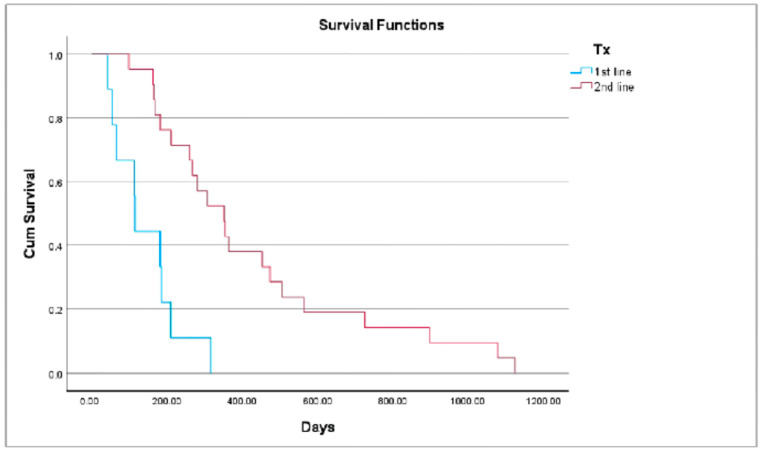
Kaplan–Meyer survival analysis for all dogs. (*p* < 0.001).

**Table 1 animals-14-03420-t001:** Dog data.

Dog	Breed, Sex, Age	Diagnosis	Dosage of Chlorambucil (mg/m^2^/Week)	Dosage of Toceranib (mg/kg Tri-Weekly)	Clinical Responses	Objective Response	ST (Days)	PFS (Days)
1	Maltese, MN, 13.9YO	Thyroid carcinoma	9.5	3.0	No	PD	351	43
2	Mongrel, FS, 10YO	Thyroid carcinoma	21.1	2.7	No	PD	452	25
3	Wirehaired fox terrier, FS, 10.5YO	Lung carcinoma	9.9	2.5	Yes	SD	113	50
4	Corgi, MN, 13.7YO	Lung carcinoma	13.4	2.1	Yes	SD	164	13
5	Poodle, MN, 11YO	Lung carcinoma	13.5	2.2	Yes	SD	1078	262
6	Jack Russell, MN, 14YO	Lung carcinoma	12.0	2.5	Yes	SD	897	382
7	Shiba inu, MN, 10.9YO	Nasal carcinoma	23.9	2.1	Yes	SD	306	98
8	Toy poodle, FS, 14.9YO	Nasal carcinoma	14.0	2.1	No	PD	353	48
9	Bichon Frise, MN, 12.5YO	Nasal carcinoma (SCC)	14.9	2.5	No	PD	L	57
10	Schnauzer, MN, 8.5YO	Frontal sinus carcinoma (SCC)	10.8	2.7	Yes	PR	210	134
11	Pekingese, FS, 11YO	Mammary carcinoma	15.1	1.9	Yes	SD	65	45
12	Poodle, FE, 6.1YO	Mammary carcinoma	19.1	2.7	No	PD	L	22
13	Poodle, FS, 12.9YO	Mammary carcinoma	11.6	2.4	No	PD	315	55
14	Mixed, FE, 12YO	Mammary carcinoma	24.8	2.8	Yes	SD	473	U
15	Mixed, FN, 10YO	Mammary carcinoma	32.0	2.8	Yes	SD	182	U
16	Poodle, FS, 13.5YO	Mammary carcinoma	15.7	2.5	Yes	SD	L	43
17	Staffordshire bull terrier, FS, 10.8YO	Mammary carcinoma (& T cell lymphoma)	20.2	2.8	Yes	SD	505	72
18	Corgi, MN, 8YO	Lingual carcinoma (SCC)	10.1	2.1	Yes	PR	98	48
19	Toy poodle, MN, 9YO	Prostatic carcinoma	19.1	2.8	Yes	SD	267	46
20	English bulldog, FS, 8.8YO	AGASACA	10.9	2.0	Yes	SD	725	283
21	Poodle, MN, 6YO	AGASACA	29.8	2.8	Yes	SD	A	42
22	Mixed, ME, 12YO	Hepatocellular carcinoma	15.2	2.8	Yes	SD	563	373
23	Boxer, ME, 5YO	Clear cell renal carcinoma	16.0	2.8	Yes	SD	363	363
24	Mongrel, FS, 12YO	MCT (low grade)	18.5	2.4	No	PD	53	18
25	Schnauzer, MN, 15.9YO	MCT (high grade)	13.7	2.2	No	PD	L	27
26	Bichon Frise, MN, 11YO	MCT	14.0	2.9	No	PD	L	20
27	Poodle, MN, 14YO	MCT	13.4	2.9	Yes	CR	1124	537
28	Poodle, FS, 11YO	MCT	12.5	2.7	No	PD	209	15
29	Golden retriever, FS, 8YO	MCT (high grade)	13.3	3.0	Yes	SD	162	154
30	Labradoodle, FS, 9.8YO	MCT (high grade)	10.9	2.9	No	PD	259	15
31	Poodle, FS, 15.2YO	MCT (high grade)	14.5	2.4	No	PD	A	17
32	Border collie, MN, 11YO	Occipital sarcoma	19.6	2.4	No	PD	114	15
33	Labrador, MN, 7YO	Fibrosarcoma	12.2	1.7	No	PD	280	48
34	Poodle mix, MN, 14YO	Liposarcoma	20.9	2.1	No	PD	181	19
35	Dachshund, FS, 6.3YO	Haemangiosarcoma	37.2	2.3	No	PD	167	22
36	Golden retriever, MN, 15.8YO	Haemangiosarcoma	16.5	2.1	No	PD	42	15
37	French bulldog, MN, 8YO	Haemangiosarcoma	20.3	2.3	Yes	SD	184	111
38	Schnauzer, FS, 11.4YO	GIST	23.8	2.5	No	PD	L	12

The data of 38 dogs treated with Toceranib and Chlorambucil, ST (survival time), PFS (progression-free survival), MN (male neutered), FS (female spayed), FE (female entire), CR (complete response), PR (partial response), SD (stable disease), PD (progressive disease), SCC (squamous cell carcinoma), A (alive), L (loss of following up), U (unavailable PFS).

**Table 2 animals-14-03420-t002:** Summarization of AEs in dogs receiving Chlorambucil and Toceranib combination (n = 38) with AEs graded by VCOG-CTCAE v2.

		G1		G2		G3		G4	G5
		Number of Dogs	Percentage of Dogs	Number of Dogs	Percentage of Dogs	Number of Dogs	Percentage of Dogs		
GIT	Vomiting	5	13.1%	0	0%	0	0%	0	0
	Diarrhea	5	13.1%	7	18.4%	0	0%	0	0
	Anorexia	4	10.5%	2	5.3%	2	5.3%	0	0
Hematological	Neutropenia	4	10.5%	0	0%	0	0%	0	0
	TCP	1	2.6%	0	0%	1	2.6%	0	0
	ALT elevation	0	0%	4	10.5%	1	2.6%	0	0
	Creatinine elevation	0	0%	2	5.3%	0	0%	0	0
Renal	High UPC ratio	2	5.3%	2	5.3%	2	5.3%	0	0

G1 (grade1), G2 (grade2), G3 (grade3), G4 (grade4), G5 (grade5), GIT (gastrointestinal), ALT (Alanine aminotransferase), TCP (thrombocytopenia), UPC (urine protein to creatinine).

## Data Availability

The data supporting this study’s findings will be available from the authors upon reasonable request.

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
