# Peer review of "Retrospective Safety Evaluation of Combined Chlorambucil and Toceranib for the Treatment of Different Solid Tumours in Dogs"

_animals, 2024, doi:10.3390/ani14233420_

Round 1
Reviewer 1 Report
Comments and Suggestions for Authors
Abstract:
Avoid the use of abbreviations which is not common to general readers (not specialist oncologist) without first introducing the abbreviation in full. Only abbreviation which will be repeated more than once can be used. Otherwise, good to spell out the full words.
Introduction
In general, please use “dog” or “canine” instead of “patient” throughout the manuscript. This is for the general reader to understand.
Do the authors know the c-kit expression or mutation status for all the tumours investigated in this study?
Line 59: worth mentioning what are the types of studies or which cancer types investigated.
Materials and methods:
Line 121: it is a bit puzzling to understand how the two referral hospitals can administer new protocols with combination drugs over the few years without any safety profiles done prior? Was there any ethical approval for this protocol to be used in patients? If it was a prospective study, then it is understandable and will still require an ethical clearance.
Line 122: solid tumour - were they removed surgically first and then the treatment was given or are all the cases involve non-resectable tumour cases? Was radiation therapy done in any of these cases which is commonly done in solid tumour cases. And if radiation therapy was done at any point of time in the specific case, would it still be included in this analysis or excluded? This should be listed under the inclusion and exclusion criteria.
Line 124: what is meant by did not have adequate follow up. please spell out the inclusion and exclusion criteria with more details.
Line 132-133: therefore, the dosages of treatment received by the different dogs in the two referral centre cohorts can have a high variation. How is that possible to draw a solid conclusion on the safety and efficacy?
Line 134: this is something no need to be mentioned. what is important is the final dosage the dog received.
Line 136: since this is a retrospective study, one cannot target the dose received by the patient. unless this is an inclusion criteria worthy mentioned.
Line 138: really do not need to mention the formulations. Since it is manufacturer's available product and most important is the final dosage received by the dog(s).
Results:
Line 177: which type of lymphoma? so this case had 2 different neoplasm and which one was included in this study?
Line 177: how do you measure response to treatment for this neoplasm GIST?
Line 180: The mammary carcinoma cases- were these all-non-resectable cases or treatment was prescribed post-surgery? What are their stages?
Line 182: normally we don’t say multiple metastasis to LN. It is sufficient to say with lymph node metastases. I wonder why this case was specifically mentioned about metastases but the other cases none? Were the other cases staged on imaging methods and LN assessment done as well?
Line 259: do you know the c-kit mutation status for this mct case?
Line 294: No statistics was done but from the KM curves you should provide the OR/ HR with confidence intervals and a pvalue regardless if it was significant or not. This is because the two groups shown in the KM curve looks decently separated and as a reader, I would be curious to know the p value.
Results section can be revised especially the paragraph before the Table2, appears to report the same data presented in the Table2. Table should be self-explanatory. This section can be significantly reduced.
Results: Please include information on how many ‘doses” of chlorambucil and toceranib phosphate each of the dogs received, can add into Table1. Or mention this information in the text- how many doses did all the dogs receive (number)? How many doses did the 1st line and 2nd line treated dogs receive in total for each of the toceranib and chlorambucil drugs.
Discussion
Please include a reason what could be the possible explanation for the first line therapy combined drug treatment resulted in shorter survival outcome compared to second line therapy as shown on the KM curves.
Lines 405-414 discussion is not relevant to the present study pertaining the combination drug effects. Do not need to elaborate more details on other combination drug protocols. There are many combination protocols described in the veterinary oncology literature.
Conclusion
The specific dosage range evaluated for the two drugs combined must be stated in the conclusion of a safety study. And whether the combined treatment as first or second line therapy is worth to be reported in the conclusion. Avoid overstatement when no PKPD of combined drugs was done and whether objective response was achieved or not is something should not be overstated since there were way too many confounding factors in this study from the various types of tumours with a variety of stages with not so clear information on whether the dogs all were staged using a CT, unknown status of c-kit and other tyrosine kinase expression in the tumours and etc.
Reference NO 33. not sure if this a typo but what type of specific reference is this? “Supporting document?”
Author Response
Abstract:
Comment 1: Avoid the use of abbreviations which is not common to general readers (not specialist oncologist) without first introducing the abbreviation in full. Only abbreviation which will be repeated more than once can be used. Otherwise, good to spell out the full words.
Response 1: Thank you for pointing this out. Modifications have been made to our revised manuscript.
Introduction:
Comment 2: In general, please use “dog” or “canine” instead of “patient” throughout the manuscript. This is for the general reader to understand.
Response 2: Thank you for the kind advice, as you suggested “patient” has been removed from the revised manuscript and changed to dog or canine throughout the manuscript.
Comment 3: Do the authors know the c-kit expression or mutation status for all the tumours investigated in this study?
Response 3: Thank you for raising this question, we acknowledge that it could be interesting to know the c-KIT status. However, most oncologists no longer test for c-Kit because of limited clinical benefits, so due to the retrospective nature of this study, this information is not available.
Comment 4: Line 59: worth mentioning what are the types of studies or which cancer types investigated.
Response 4: Thank you for the comment. We have added the information of the study types and cancer types included accordingly to improve the comprehensiveness of the paragraph.
Materials and methods:
Comment 5: Line 121: it is a bit puzzling to understand how the two referral hospitals can administer new protocols with combination drugs over the few years without any safety profiles done prior. Was there any ethical approval for this protocol to be used in patients? If it was a prospective study, then it is understandable and will still require an ethical clearance.
Response 5: Thank you for your comment, we appreciate and understand your concerns. This is a retrospective study, ethical approval is stated at the bottom of the manuscript. There are many studies assessing the safety profile of toceranib and chlorambucil as a sole treatment or in combination with other drugs. The novelty of our study is the combination of chlorambucil and toceranib together. The choice of both drugs is quite rational, and other oncologists are likely to have used them in combination a few times during their clinical practice. However, until now, nobody has reported this, hence this preliminary study of which the results will need to be confirmed by a prospective clinical study, that has been already planned.
Comment 6: Line 122: solid tumour - were they removed surgically first and then the treatment was given or are all the cases involve non-resectable tumour cases? Was radiation therapy done in any of these cases which is commonly done in solid tumour cases. And if radiation therapy was done at any point of time in the specific case, would it still be included in this analysis or excluded? This should be listed under the inclusion and exclusion criteria.
Response 6: Thank you for raising the key point for improving the details for our inclusion and exclusion criteria. Of the 38 dogs included in our study, 19 of them were non-resectable tumour cases, and the other 9 cases had surgery before chemotherapy. None of the cases had received radiotherapy. We added this in the text.
Comment 7: Line 124: what is meant by did not have adequate follow up. please spell out the inclusion and exclusion criteria with more details.
Response 7: Thank you for the follow-up comment regarding our inclusion and exclusion criteria. We have added more information regarding this issue in the revised text.
Comment 8: Line 132-133: therefore, the dosages of treatment received by the different dogs in the two referral centre cohorts can have a high variation. How is that possible to draw a solid conclusion on the safety and efficacy?
Response 8: Thank you for your comment, considering the retrospective nature of the study, the different treatment goals for each patient, the different sizes of the dog and the only available 2mg oral tablet formulation, it is expected that dosage was variable from patient to patient. As you pointed out, this is for sure the main limitation of the study, however, our data gives important information of the safety profile of this drug combination at the dose range that was used in our study and these data give a good starting point for future prospective studies with dose escalation. We had made this limitation clear in the text.
Comment 9 Line 134: this is something no need to be mentioned. what is important is the final dosage the dog received.
Response 9: Thank you for the feedback. We agree that the final dosage the dog received is the point we are focusing on. We included this sentence to explain that the available tablets and the size of the dogs are the main factors that affected the final dosages.
Comment 10: Line 136: since this is a retrospective study, one cannot target the dose received by the patient. unless this is an inclusion criteria worthy mentioned.
Response 10: Thank you for pointing out this crucial point. We understand we cannot target the dose due to the retrospective nature of the study. However, as two of our authors were the clinicians who decided the dosages for the dogs in the past, the targeted dose we mentioned was the dose they were aiming at that time based on the available literature on the safety profile of each drug. We have updated this paragraph and removed the target dosage to avoid confusion.
Comment 11: Line 138: really do not need to mention the formulations. Since it is manufacturer's available product and most important is the final dosage received by the dog(s).
Response 11: Thank you for the comment. We have made changes according to our revised manuscript. We believe it is better to mention what formulation of the drug was used as compounding drugs are possible but are proven to not be completely reliable. The tablet used also justify the different schedules based on the size of the dog.
Results:
Comment 12: Line 177: which type of lymphoma? so this case had 2 different neoplasms and which one was included in this study?
Response 12: Thank you for the question. This dog was diagnosed with T cell lymphoma and mammary carcinoma. It was included as a mammary carcinoma case. This dog was on chlorambucil for low-grade T-cell lymphoma initially and toceranib was added when it developed metastatic mammary carcinoma.
Comment 13: Line 177: how do you measure response to treatment for this neoplasm GIST?
Response 13: Thank you for your comment. The response was evaluated by ultrasound as it was an intestinal tumour.
Comment 14: Line 180: The mammary carcinoma cases- were these all-non-resectable cases or treatment was prescribed post-surgery? What are their stages?
Response 14: Thank you for the question. There were three cases of stage II mammary carcinoma, one case of stage III mammary carcinoma, and two cases of stage V mammary carcinoma. For the two stage II cases, they were recurred tumours after surgery, with widespread satellite cutaneous nodules (inflammatory carcinoma) and hence started the chemotherapy. The third stage II case had also low-grade T cell lymphoma. This dog was on chlorambucil for low-grade T-cell lymphoma initially and toceranib was added when it developed metastatic mammary carcinoma. For the stage III case, the owner opted for chemotherapy as the initial treatment. For the two stage V cases, they had lung metastases and were only treated with chemotherapy. This has been specified in the revised manuscript.
Comment 15: Line 182: normally we don’t say multiple metastasis to LN. It is sufficient to say with lymph node metastases. I wonder why this case was specifically mentioned about metastases but the other cases none? Were the other cases staged on imaging methods and LN assessment done as well?
Response 15: Thank you for pointing this mistake out. We agree that there is no reason to specifically mention this case and we have combined this specific case with the other mammary carcinoma cases, making in total six cases of mammary carcinoma. All the cases were staged with imaging and LN assessment under owner consent.
Comment 16: Line 259: do you know the c-kit mutation status for this mct case?
Response 16: Thank you for the question. According to the available data, unfortunately, we do not know the c-kit mutation status for this MCT case.
Comment 17: Line 294: No statistics was done but from the KM curves you should provide the OR/ HR with confidence intervals and a pvalue regardless if it was significant or not. This is because the two groups shown in the KM curve looks decently separated and as a reader, I would be curious to know the p value.
Response 17: Thank you for the comment. We included the p-value in the KM curves. We have added it in the text as well for more clarity. Regarding our study, we did not do COX or other regression analysis due to the low number of cases and different confounding factors (small number of cases in each cancer group, different stages etc). Our statistics are mainly descriptive as mentioned in our manuscript and we presented the KM curve for reader to just have an idea of different survivals, this can be even removed if necessary, as this is not the main aim of our study.
Comment 18: Results section can be revised especially the paragraph before the Table2, appears to report the same data presented in the Table2. Table should be self-explanatory. This section can be significantly reduced.
Response 18: Thank you for the recommendation. We agree the result section should be revised and avoid repeated information from paragraphs and tables. We have made changes accordingly to keep the results delivered in a clearer manner.
Comment 19: Results: Please include information on how many ‘doses” of chlorambucil and toceranib phosphate each of the dogs received, can add into Table1. Or mention this information in the text- how many doses did all the dogs receive (number)? How many doses did the 1st line and 2nd line treated dogs receive in total for each of the toceranib and chlorambucil drugs.
Response 19: Thank you for the comment. We have added the dosages of Chlorambucil and Toceranib each dog received into Table 1 for a higher transparency of our data.
Discussion:
Comment 20: Please include a reason what could be the possible explanation for the first line therapy combined drug treatment resulted in shorter survival outcome compared to second-line therapy as shown on the KM curves.
Response 20: Thank you for your comment. We also noted this unexpected finding of dogs receiving second-line therapy, in general, have longer survival and we agree we should discuss this result. One of the potential explanations could be the clinician bias, this combination was often used in advanced cases with poor life expectations. While patients who received other drugs before starting the combination, could have a better prognosis from the first diagnosis. Thank you for the suggestion. We added a paragraph at line 396-401 in the discussion section.
Comment 21: Lines 405-414 discussion is not relevant to the present study pertaining the combination drug effects. Do not need to elaborate more details on other combination drug protocols. There are many combination protocols described in the veterinary oncology literature.
Response 21: Thank you for your comment, we agree and removed this part form the revised text
Conclusion:
Comment 22: The specific dosage range evaluated for the two drugs combined must be stated in the conclusion of a safety study. And whether the combined treatment as first or second line therapy is worth to be reported in the conclusion. Avoid overstatement when no PKPD of combined drugs was done and whether objective response was achieved or not is something should not be overstated since there were way too many confounding factors in this study from the various types of tumours with a variety of stages with not so clear information on whether the dogs all were staged using a CT, unknown status of c-kit and other tyrosine kinase expression in the tumours and etc.
Response 22: Thank you for the comment. We agree that we should be careful with our words of choice to draw a conclusion to avoid overstatement. We have made changes in our revised manuscript.
Comment 23: Reference NO 33. not sure if this a typo but what type of specific reference is this? “Supporting document?”
Response 23: Thank you for noticing our incorrect reference format. We have updated the reference list to remove the error.
Reviewer 2 Report
Comments and Suggestions for Authors
This retrospective study assesses the safety and clinical efficacy of combining Chlorambucil and Toceranib in treating various solid tumors in dogs. Thirty-eight canine patients received this combination therapy, with adverse events (AEs) primarily being mild to moderate, including gastrointestinal, hematological, and renal issues. The clinical benefit rate reached 55.3%, with no treatment discontinuations due to AEs. This suggests that the combination is well-tolerated, though more extensive prospective trials are needed to confirm efficacy and investigate therapeutic potential across tumor types. The findings highlight a potentially synergistic anti-angiogenic effect, indicating promising applications for treating canine tumors with this drug combination.
However, some questions need to be further clarified:
1. Were informed consent procedures used for all patient data, especially given the retrospective design?
2. Was there a rationale for selecting the sample size of 38 dogs, or was it based solely on available data?
3. Could more detailed inclusion and exclusion criteria be added to enhance transparency, especially for patients with comorbidities?
4. Could more specific details about the dosing adjustments based on body size be provided?
5. Were any controls used in this retrospective study to minimize potential selection bias?
6. Would it be feasible to include a comparison group treated with only one of the two drugs to highlight the combination's impact better?
7. Can the authors clarify if any specific breed had a higher incidence of AEs? This would be useful for breed-specific considerations in future studies.
8. How was the “clinical benefit rate” calculated, and could this be elaborated upon for better clarity?
9. Given the variability in tumor types, would it be beneficial to analyze AE incidence in sub-groups based on tumor type?
10. The study mentions that combined therapy may have a synergistic anti-angiogenic effect. Would the authors consider discussing potential cellular mechanisms, even if speculative, to support this claim?
11. Were there any observed trends in PFS or survival times among different tumor types that could help identify the most responsive cancer types?
12. Given that only retrospective data is available, how would the authors suggest confirming efficacy in a prospective clinical trial?
13. Could the authors outline specific recommendations for clinicians interested in this combination therapy? Practical guidance would enhance the paper's real-world applicability.
Please incorporate these answers in the MM and discussion sections.
Author Response
Comment 1: Were informed consent procedures used for all patient data, especially given the retrospective design?
Response 1: Thank you for the question. Yes, the informed consent procedures were used for all patient data.
Comment 2: Was there a rationale for selecting the sample size of 38 dogs, or was it based solely on available data?
Response 2: Thank you for the question. The 38 dogs were selected as they were the only available cases using the combination of Chlorambucil and Toceranib.
Comment 3: Could more detailed inclusion and exclusion criteria be added to enhance transparency, especially for patients with comorbidities?
Response 3: Thank you for the kind advice. We have elaborated more details on the inclusion and exclusion criteria in the revised manuscript.
Comment 4: Could more specific details about the dosing adjustments based on body size be provided?
Response 4: Thank you for your comment, we specified in the text that the adjustment of dosing was mainly based on the different sizes of the dogs and the available oral tablet formulation, but being retrospective, many factors come to play when choosing the dose (owner risk tolerance, finance, type of cancer and treatment goal).
Comment 5-6: Were any controls used in this retrospective study to minimize potential selection bias? Would it be feasible to include a comparison group treated with only one of the two drugs to highlight the combination's impact better?
Response 5-6: Thank you for raising the question about controls. Unfortunately, we did not have any controls for this study. However, the main aim of the study is to assess the adverse events of this drug combination, and despite not having control patients there are extensively published data on the side effects of toceranib and chlorambucil as a single agent.
Comment 7: Can the authors clarify if any specific breed had a higher incidence of AEs? This would be useful for breed-specific considerations in future studies.
Response 7: Thank you for the suggestion. As we do not have an even distribution for the breeds of all our included dogs, in which most of them were poodles, we cannot draw any valid conclusion on which breed having a higher incidence of AEs.
Comment 8: How was the “clinical benefit rate” calculated, and could this be elaborated upon for better clarity?
Response 8: Thank you for the comment. The calculation method for “clinical benefit rate” can be found in line 166-167 “while clinical benefit rate (CBR) was calculated by summation of dogs achieving CR, PR, and SD for at least six weeks.”
Comment 9: Given the variability in tumor types, would it be beneficial to analyze AE incidence in sub-groups based on tumor type?
Response 9: Thank you for your comment. We do not think the type of tumour can influence significantly the type or grade of adverse events, so we did not have a look in to this.
Comment 10: The study mentions that combined therapy may have a synergistic anti-angiogenic effect. Would the authors consider discussing potential cellular mechanisms, even if speculative, to support this claim?
Response 10: Thank you for your comment. we have added extra information in the text.
Comment 11: Were there any observed trends in PFS or survival times among different tumor types that could help identify the most responsive cancer types?
Response 11: Thank you for the comment. It is difficult to conclude on which cancer types responded the most to combined chemotherapy simply based on PFS or survival times as there are multiple factors affecting the PFS and survival times, including the stage of the tumour in each case, the aggressiveness nature of each tumour due to their origin or types, whether the tumour is a recurred case or not, and whether the dog has other unrelated diseases or not. However, we noticed a higher response rate (as expected) in MCT.
Comment 12: Given that only retrospective data is available, how would the authors suggest confirming efficacy in a prospective clinical trial?
Response 12: Thank you for the comment. As mentioned in line 437, we recommend conducting larger prospective studies on specific types of cancers with escalating dosages of chlorambucil.
Comment 13: Could the authors outline specific recommendations for clinicians interested in this combination therapy? Practical guidance would enhance the paper's real-world applicability.
Response 13: Thank you for your comments, we believe that the combination at the dosage reported is very safe. This combination could be potentially more effective than toceranib alone in all the tumours already known to benefit from toceranib alone.
Round 2
Reviewer 1 Report
Comments and Suggestions for Authors
the manuscript has been edited and improved compared to the first submission. The authors have addressed most of the main issues raised and corrected errors where necessary.